# Enhancing single-cell transcriptomics using interposed anchor oligonucleotide sequences
Jianfeng Sun [1], Martin Philpott[1], Danson Loi[1], Gabriela Hoffman[2], Jonathan Robson[2], Neelam Mehta[1], Eleanor Calcutt[1], Vicki Gamble[1], Tom Brown Jr[2], Tom Brown Sr [3], Udo Oppermann[1,4] & Adam P. Cribbs [1,4] ✉

Single-cell transcriptomics, which utilises barcodes and unique molecular identifiers (UMIs) for polyA+ mRNA capture, is compromised by oligonucleotide synthesis errors. To address this, we modified the oligonucleotide capture design and integrated an interposed anchor between the barcode and the UMI. This design significantly reduces the need to discard reads due to synthesis inaccuracies. Our results demonstrate that this anchor-enhanced design substantially improves gene expression profiles in droplet-based single-cell sequencing analyses.

Droplet-based sequencing has rapidly advanced to be the gold standard method in studying single-cell transcriptomics by leveraging its capability to offer higher throughput. Droplet-based sequencing methods such as Drop-seq[1], InDrops[2] and 10x Chromium[3] provide improved throughput at a reduced cost per cell.

Central to these technologies are oligonucleotide sequences that are synthesised and contain a polyA+ capture region, a PCR primer, a cell barcode and a unique molecular identifier (UMI). Historically, the techniques and strategies underpinning droplet-based sequencing were developed with Illumina sequencing platforms in mind. Typically, the Illumina sequencing protocol employs a fragmentation step so that sequencing of Read 1 contains both the barcode and the UMI sequence, while, Read 2 contains the sequencing information for either the 3' or 5' end of the captured transcript. However, the emergence of long-read sequencing, which allows for complete end-to-end sequencing, has made identification of the barcode and UMI sequences more challenging. This stems from the reliance on computational pattern-matching to locate the primer site upstream of the barcode and UMI. PCR and sequencing errors further complicate this[4,5], making barcode and UMI identification less accurate and leading to a significant proportion of reads being discarded.

Two primary strategies are employed for the synthesis of oligonucleotides in droplet-based sequencing: on-bead chemical synthesis and enzymatic ligation. The on-bead chemical method, pioneered by Drop-seq, utilises a split and pool technique to generate barcodes and synthesises random UMIs from single random nucleosides, effectively creating diverse barcodes and ensuring most cells align with a bead. In contrast, the enzymatic ligation method, introduced by InDrops and subsequently adopted by other techniques, employs combinations of pre-synthesised oligonucleotides, where barcodes are generated using a split and pool approach with small pre-synthesised building blocks, and random UMIs are added to the barcode through enzymatic ligation. Despite these different methodologies, both strategies suffer from a significant drawback: substantial truncation due to difficulties in purifying oligonucleotides once affixed to the beads, as immobilised oligonucleotides cannot be cleaved and purified because the cell barcodes are integral to each sequence. While oligonucleotides synthesised by commercial providers are of high quality with purification options (e.g., PAGE or HPLC) for truncated oligonucleotides, this purification is not feasible for single-cell droplet-based sequencing due to the attachment of oligos to beads. The field has not fully recognised the consequences of truncated oligonucleotides on downstream analysis.

In previous work, we introduced homodimer and homotrimer UMIs to counteract PCR and sequencing inaccuracies across both bulk and single-cell resolutions[4,5]. In this study, we identify another critical source of error: oligonucleotide synthesis inaccuracies. To address this, we have developed a bead design that alleviates these synthesis issues. Crucially, our design aligns with both short-read and long-read sequencing methods. Our findings suggest that anchors incorporated into the oligonucleotide capture sequence before and after the UMI can enhance accurate barcode and UMI identification. These adjustments have resulted in a marked improvement in UMI recovery and a heightened feature detection rate, improving the capabilities of droplet-based sequencing.

[1]Botnar Research Centre, Nuffield Department of Orthopaedics, Rheumatology and Musculoskeletal Sciences, National Institute of Health Research Oxford Biomedical Research Unit (BRU), University of Oxford, Oxford, UK. [2]ATDBio Ltd (now part of Biotage), Oxford, UK. [3]Chemistry Research Laboratory, Department of Chemistry, University of Oxford, Oxford, UK. [4]Oxford Centre for Translational Myeloma Research University of Oxford, Oxford, UK.
✉e-mail: adam.cribbs@ndorms.ox.ac.uk

## Results

### Evidence of bead truncation in 10x chromium and Drop-seq datasets

Droplet-based single-cell methods, such as Drop-seq and 10x Chromium, employ oil droplets to individually encapsulate and process mRNA from cells in a highly parallel manner. In this process, cells are sequestered within droplets, enabling cell lysis and subsequent mRNA capture via oligonucleotide-coated beads. The designs of these beads, however, exhibit notable differences (Fig. 1a). In Drop-seq, the bead features a PCR primer region followed by a 12-base pair (bp) cell barcode, created by a split and pool synthesis. This is succeeded by an 8 bp Unique Molecular Identifier (UMI) sequence that includes a V (A, C, or G) base preceding the start of a poly(dT) capture region[1]. This design facilitates the linking of the barcode and UMI with the polyadenylated mRNA. Conversely, the bead design used in the 10x Chromium system includes a 16 bp barcode produced by combinatorial enzymatic ligation and split and pool synthesis, coupled with a 12 bp UMI sequence. Unlike Drop-seq, the 10x Chromium beads lack a V base between the UMI and the poly(dT) sequence[3]. They uniquely incorporate a V base followed by an N (A, C, G or T) base at the poly(dT) end, ensuring mRNA capture near the polyA terminus of the mRNA.

Our analysis of publicly available data from 10x Chromium and Drop-seq reveals distinct nucleotide distribution patterns in Read 1, indicative of the differing bead designs and their impact on mRNA capture and sequencing. Notably, there is an elevated occurrence of the thymine (T) base in 10x Chromium data, particularly in the final position of the UMI region (Fig. 1b and Supplementary Figs. 1, 2, 3). This trend was even more pronounced in 10x Chromium Oxford Nanopore Technologies (ONT) long-read sequencing data (Fig. 1c), suggesting that sequencing extends into the poly(dT) capture region. Drop-seq[1] shows a similar increase in the proportion of T at the final UMI base but a different proportion of nucleotides across the length of the UMI (Fig. 1d). There is no equivalent long-read data for this library so a direct comparison with ONT could not be made. Overall, our analysis underscores variations in the distribution of nucleotide bases along the lengths of UMIs, indicating potential UMI bias. Notably, there is a pronounced increase in thymine (T) nucleotide occurrence towards the end of the UMI. This pattern is indicative of sequencing extending into the poly(dT) region, suggesting that the oligonucleotides may be truncated.

### Bead truncation results in diminished UMI complexity through T-base overrepresentation, but has minimal impact on cell identification

Having identified potential biases in nucleotide composition in both 10x Chromium and Drop-seq single-cell barcode and UMIs, we subsequently explored these findings in greater detail, specifically examining the issue of bead truncation. High accuracy in oligonucleotide synthesis is essential for successful single-cell sequencing, as errors can lead to incorrect barcode matching and inflated count matrices due to misidentified UMIs. Despite the prevalent adoption of solid-phase phosphoramidite oligonucleotide synthesis, the method attains approximately 99% coupling efficiency per cycle[6,7]. Consequently, the proportion of full-length oligonucleotides with the correct sequence diminishes markedly with increasing length. For instance, a synthesised strand comprising 100 nucleotides will typically exhibit less than 50% of the molecules containing the intended sequence[8,9].

To explore the occurrence and nature of synthesis errors, we sequenced 10x Chromium and Drop-seq oligonucleotides in isolation. This process revealed a predictable predominant peak size of 28 bp for the 10x Chromium beads (Fig. 2a) and 20 bp for Drop-seq (Fig. 2b). Nonetheless, a notable truncation was observed, with merely 43.5% of the 10x Chromium beads and 35% of the Drop-seq beads exhibiting the anticipated length, demonstrating bead truncation. Subsequently, we analysed the counts for the top 20 UMIs in both 10x and Drop-seq experiments (Fig. 2c, d), uncovering a consistent trend of T-base enrichment, especially at the end of UMIs. This further substantiates the occurrence of bead truncation.

To elucidate the impact of potential synthesis inaccuracies on barcode assignment, we conducted a single-cell species-mixing experiment using both the 10x Chromium and Drop-seq methodologies. For both methods, the barcode sequences were computationally truncated. Our results indicate that truncation does not adversely affect the quantification of cells identified (Supplementary Fig. 4). This outcome is attributed to the efficacy of the whitelisting correction strategy, which is capable of overcoming errors or truncations. This underscores the resilience and reliability of the whitelisting

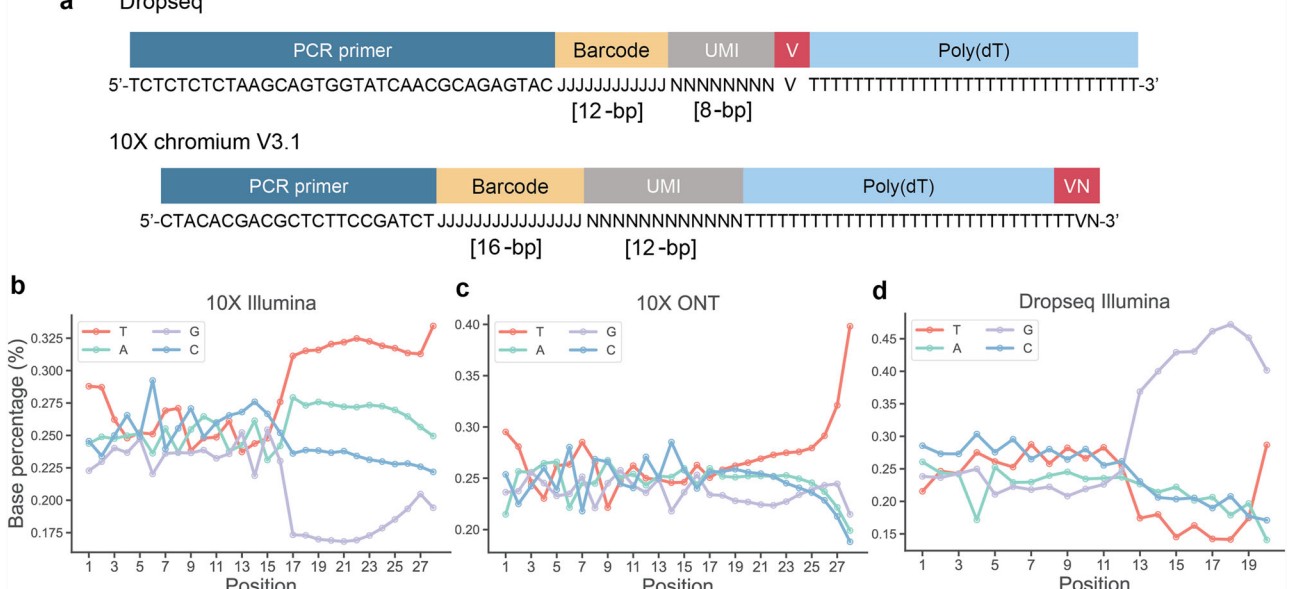

**Fig. 1 | Analysis of the base composition of 10x Chromium and Drop-seq barcodes and UMIs. a** A comparative schematic illustrates the bead designs of both Drop-seq and 10x Chromium platform v3.1. While both designs incorporate a barcode and UMI, they vary in length, the positioning of the V base (A, C, or G), and PCR primer sequences. N signifies a random base, J denotes a semi-random base, and T, C, G and A represent Thymidine, Cytosine, Guanine, Adenosine, respectively. **b** The 10x Chromium 5k PBMC v3.1 dataset was downloaded, and analysis was performed for short-read Illumina sequencing. **c** The 10x Chromium 5k PBMC v3.1 dataset was downloaded, and analysis was performed for long-read ONT sequencing. **d** Illumina sequencing data from Macosko et al.[1] was downloaded and analysed. The proportion of each base is plotted across the barcode and UMI sequences.

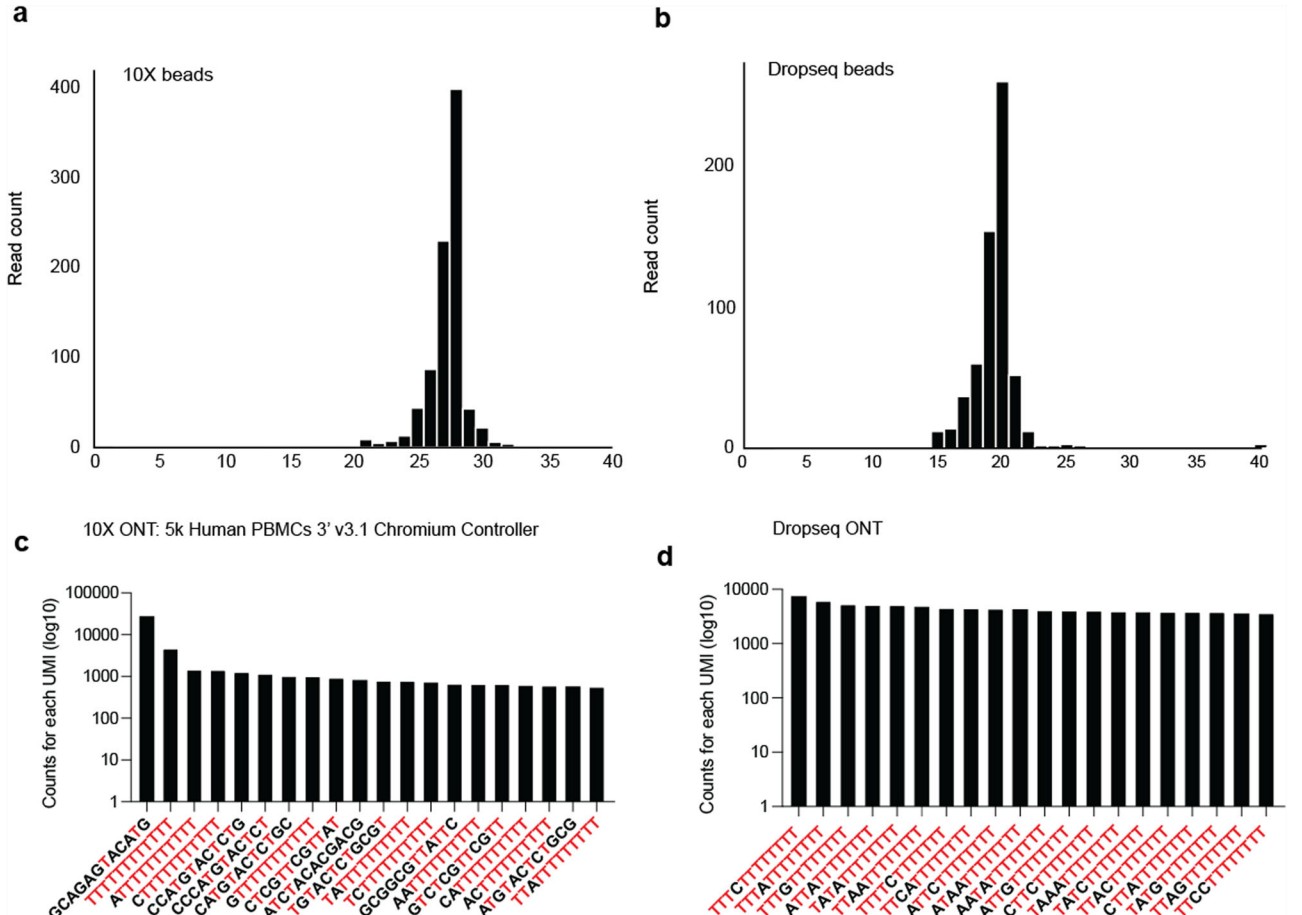

**Fig. 2 | Impact of bead truncation on UMI deduplication accuracy. a** Generation of a sequencing library from 10x bead oligos and then sequenced using ONT flongle. **b** A library was generated from oligos on Drop-seq beads and then sequenced using ONT flongle. **c** Visualisation of the top 20 overrepresented UMIs in the 10x Chromium 5k PBMC v3.1 dataset, displaying the count of each UMI. **d** Visualisation of the top 20 overrepresented UMIs in the Macosko et al.[1] dataset, displaying the number of UMI counts.

correction mechanism in maintaining accurate cell identification, despite potential synthesis errors.

### UMIs are significantly impacted by synthesis errors, and incorporating an anchor improves detection accuracy

Considering the intrinsic randomness of UMIs, error correction cannot be effectively implemented through whitelisting techniques. Consequently, errors associated with UMIs can lead to an overestimation of read counts, potentially impacting downstream analyses such as differential gene expression. Our investigations have identified evidence of truncation in the oligonucleotides synthesised on beads, underscoring the challenges in achieving accurate UMI-based quantification (Figs. 1 and Fig. 2; Supplementary Fig. 1). Truncating UMIs computationally by one base led to 115 differentially expressed transcripts between 11 and 12-base UMIs (Supplementary Fig. 5 and Supplementary Data **1**-Table 1). Differential expression varied across cell types (Supplementary Fig. 6), indicating that UMI truncation may compromise gene expression accuracy. Our findings highlight that while barcode accuracy remains largely unaffected, truncations in UMI sequences can significantly impact the reliability of gene expression quantification.

Having identified synthesis challenges associated with the bead oligonucleotides, we theorised that incorporating an anchor sequence (BAGC) between the barcode and UMI, and a V base between the UMI and the poly(dT) capture handle, could provide clearer demarcation of the beginning of the UMI. In our previous work[10], we introduced the concept of a Common Molecular Identifier (CMI) sequence, designed to precisely quantify errors in a sequenced read (Supplementary Fig. 7). In the present study, we employed the CMI to evaluate the merits of incorporating an anchor within the oligonucleotide. To this end, we synthesised a free oligonucleotide that contained a PCR handle, a constant 12 bp barcode region, preceded by a 4 bp anchor, a 32 bp homodimer CMI, and finally a V base prior to the poly(dT) region (Fig. 3a). To pinpoint the CMI, we adopted two contrasting techniques. The first, known as the Positional Strategy, involved locating the end of the PCR handle through sequence alignment, then projecting the CMI's onset to be 16 base pairs distant from this terminus. Conversely, the anchor strategy discerned the start of the CMI by pattern-matching the anchor sequence, thus identifying the CMI's initiation immediately post the anchor position.

Based on the bead design, we conducted an initial computational analysis to assess the efficiency of capturing CMIs using simulated data (Fig. 3b, c and Supplementary Fig. 8). Benchmarking simulations indicated that while both the anchor and positional strategies initially highlighted similar effectiveness in CMI identification, the anchor strategy outperformed the positional strategy as error rates increased. Subsequently, experimental evaluation using an mRNA bulk library followed by ONT sequencing revealed that the anchor method significantly improved the accuracy of CMI identification compared to the positional method (Fig. 3d), demonstrating that an anchor enhances precision in determining the CMI's initial start position.

### A new bead design containing an anchor mitigates oligo truncation and coupling errors

After demonstrating that incorporating an anchor between the barcode and UMI enhances CMI identification, we integrated this feature into our Drop-

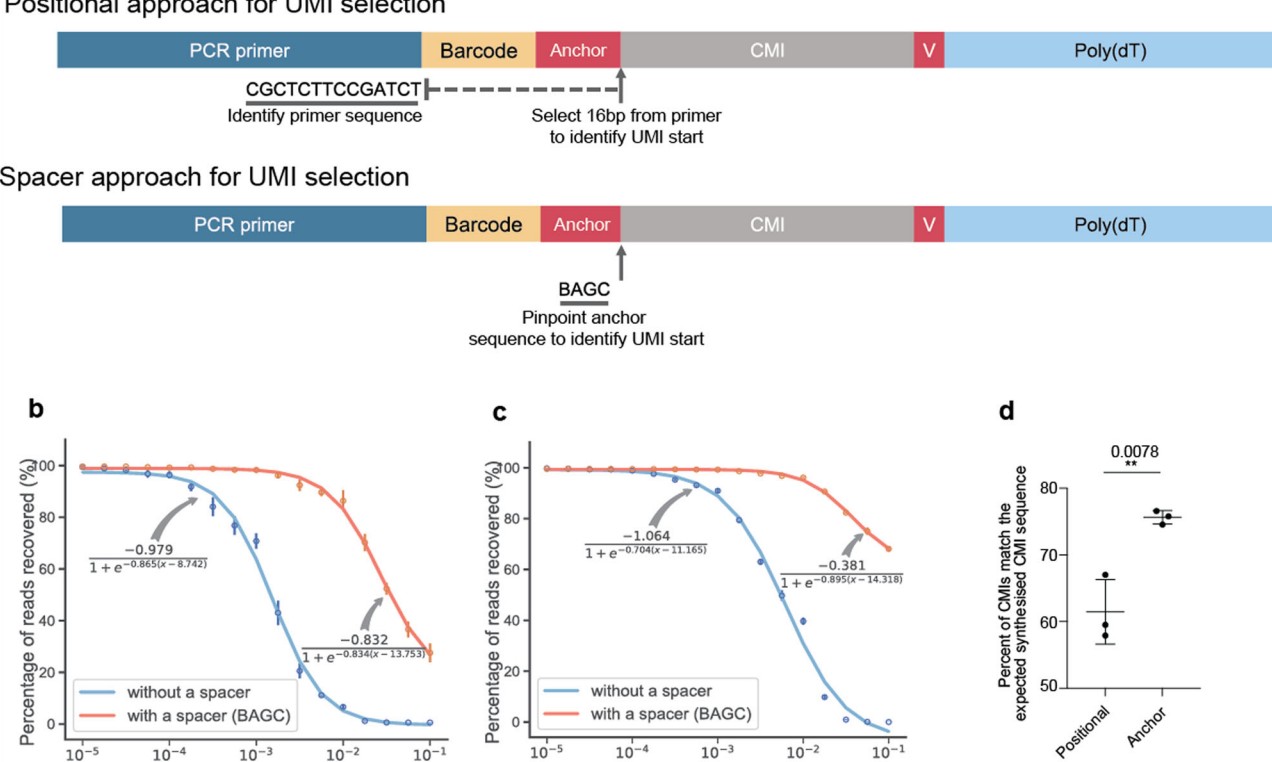

**Fig. 3 | Inclusion of an anchor enhances the ability to identify the start of the CMI sequence. a** Schematic representation comparing two CMI selection methods: positional and anchor. The positional method identifies the PCR primer's end through local alignment and selects a site 16 base pairs downstream. In contrast, the anchor method utilises a regular expression to pinpoint the CMI's start position. **b** The numbers of reads recovered were plotted as a function of the PCR error rate. **c** The numbers of reads recovered were plotted as a function of the sequencing error rate. **d** A bulk reverse transcription and PCR were performed using the oligonucleotide design in **a**. Three replicates are shown for each condition. The percentages of CMIs that perfectly matched the anticipated CMI sequence were measured for both the positional and anchor strategy. Data analysed using Unpaired *t*-test assuming both populations have the same standard deviation. **P < 0.01. The mean and S.D. are plotted.

seq bead designs. Previously, we developed an advanced Drop-seq method named single-cell corrected long-read sequencing (scCOLOR-seq)[5,11], which employs error-correcting homodimer UMIs. In the design of the capture oligonucleotide, we incorporated an anchor sequence (BAGC) upstream of the UMI and a V base downstream of the UMI (Fig. 4a). The inclusion of an anchor clearly demarked the boundary between the barcode and the UMI sequence (Fig. 4b). Our primary objective was to assess whether this modification increased the accuracy of UMI sequence detection.

Given the UMI's unique homodimer composition, we theorised that assessing perfect dimer nucleotide concordance throughout the UMI would be an effective validation metric. Improved concordance in homodimer pairs would indicate enhanced accuracy in UMI identification. Following this, we conducted a comparative analysis of the positional and anchor-based approaches to further refine our understanding of UMI characterisation. The addition of the anchor indeed increased the homodimer concordance rate by 13% to 73% (Fig. 4c), underscoring our ability to precisely determine the UMI boundaries. This marked rise in the dimer concordance rate for the homodimer UMI indicated that the anchor's inclusion improves the UMI sequence identification process.

### The inclusion of an anchor improves UMI counts and number of transcripts detected

We next tested the new bead design in a scCOLOR-seq species-mixing experiment followed by ONT sequencing. During the subsequent data analysis, we compared both the positional and anchor techniques (Supplementary Fig. 9). Notably, implementing the anchor-centric method resulted in a significant increase in the number of detected UMIs (Supplementary Fig. 10a) and transcripts (Fig. 5a and Supplementary Fig. 10b) per cell relative to the positional strategy. The anchor method shows a higher proportion of cells expressing a larger number of transcripts (Fig. 5b), suggesting increased sensitivity and detection capability when using the anchor approach. Furthermore, the greater number of features were identified within both human (Fig. 5c) and mouse (Fig. 5d, e, f) cells using the anchor approach when compared to the positional approach. This suggests that integrating an anchor in the bead oligonucleotide sequence effectively minimises artefacts, yielding a more precise tally of unique molecules across a broad spectrum of features.

## Discussion

Single-cell RNA-sequencing (scRNA-seq) technology is a rapidly advancing technology that is revealing new scientific findings[12,13]. However, the data it generates can often contain numerous technical biases[14]. In this study, we evaluated bead synthesis errors in single-cell sequencing. We identified several technical inaccuracies arising from the imperfect synthesis of bead-bound oligonucleotides. To address these errors, we devised an enhanced bead design, which improved the barcode and CMI assignment. One of the pressing issues with scRNA-seq data is the amplification of technical errors, which introduces noise[15,16], making gene counting unreliable and potentially skewing differential gene expression results. The ideal scenario would see

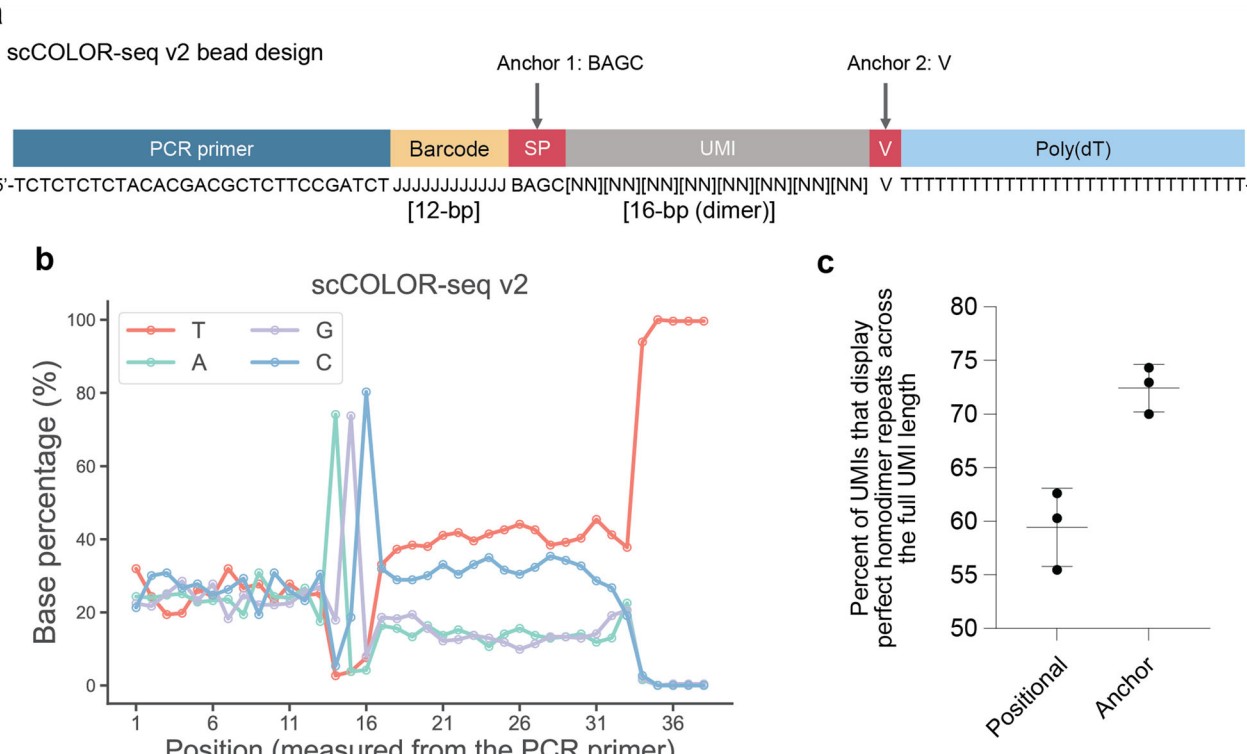

**Fig. 4 | The inclusion of an anchor in the mRNA capture beads improves UMI recovery. a** A schematic showing the scCOLOR-seq v2 bead design that incorporates an anchor sequence between the barcode and UMI, in addition to a homodimer UMI followed by a V base between the poly(dT) capture region. **b** The percent of A, C, G and T-base plotted for each base within the fastq file upstream of the PCR primer end position. **c** The homodimer repeats were used as a proxy for measuring improved UMI selection. Three replicates are shown for each condition. The percent of perfect homodimer repeats across the full length of the UMI is plotted using the positional and the anchor approach.

scRNA-seq tools accurately gauging and correcting uncertainties from these biases and errors[17,18]. In earlier studies we have identified sequencing and PCR errors as sources of technical inconsistencies in single-cell transcriptomics[5,10], causing inaccurate feature counting. Yet, the impact of oligonucleotide bead synthesis on droplet-based single-cell sequencing has been largely unexplored.

There are two primary bead synthesis strategies: on-bead chemical synthesis and enzymatic ligation. The on-bead chemical method incorporates a barcode generated using a split and pool technique, and then a random UMI synthesised using single random nucleosides, a concept first introduced by Drop-seq[1]. Its efficient creation of capturing oligonucleotides on beads offers a diverse barcode size and ensures most cells align with a bead. Conversely, the enzymatic ligation method, initially introduced by InDrops[2] and adopted by other techniques[3,19], presents a more modular approach to bead synthesis. In this strategy, bead fabrication utilises combinations of a limited set of pre-synthesised oligonucleotides. In alignment with the chemical synthesis method's principles, the barcode is created using a split and pool approach using small stretches of pre-synthesised building blocks. However, a distinct feature here is that a pre-synthesised random UMI is enzymatically ligated to the barcode. While these methods differ in their approach to bead synthesis, both exhibit a notable drawback. Our findings demonstrate that both components are susceptible to considerable truncation, attributable to incomplete coupling during chemical synthesis and the inefficacy of purification processes for enzymatic ligation precursors.

We have shown that synthesis issues significantly influence the precision of single-cell sequencing. Oligonucleotide synthesis errors, with an average rate of 1 in 100 bases[20,21], commonly result in truncated or elongated oligonucleotides. While standard oligonucleotide synthesis allows for the separation and purification of truncated sequences from full-length products using HPLC or PAGE, these methods are not possible for single-cell sequencing applications. As a result, these length discrepancies can interfere

with barcode and UMI sequence detection, a problem further amplified in long-read sequencing. This is because, in long-read sequencing, barcodes are discerned following computational alignment to the PCR primer, coupled with positional matching to pinpoint the cell barcode and UMI's beginning[22,23]. The heterogeneity in oligonucleotide lengths on beads increases the propensity for errors in detecting barcodes and UMIs. UMIs pose a greater challenge compared to barcodes; although whitelisting can rectify the majority of barcode inaccuracies, the intrinsic randomness of UMIs renders whitelisting unfeasible. To tackle synthesis-related challenges with UMI detection, we incorporated anchors and synthesised our UMIs using homodimer nucleoside amidites. This strategy not only pinpoints the start of the UMI but also identifies and fixes issues in the UMI arising from PCR, sequencing, and synthesis. As the field of single-cell biology advances and its techniques find application in clinical settings[12,24,25], the development of more rigorous methods to ensure the reliability of single-cell methodologies becomes imperative. Our findings demonstrate that the incorporation of spacers mitigates synthesis-related errors, thereby enhancing the accuracy of differential expression analysis.

## Methods
### Cell lines and reagents
Jurkat cells were purchased from ATCC. 5TGM1 was a kind gift from Prof Clair Edwards. Both Jurkat and 5TGM1 cells were cultured in a complete RPMI medium supplemented with 10% Foetal Calf Serum. All parental cell lines were tested twice per year for mycoplasma contamination and authenticated by STR during this project.

### Oligonucleotide synthesis
Single-cell oligonucleotide bead synthesis was executed in line with previously established methods[5,11], but with the following alterations to the bead design:

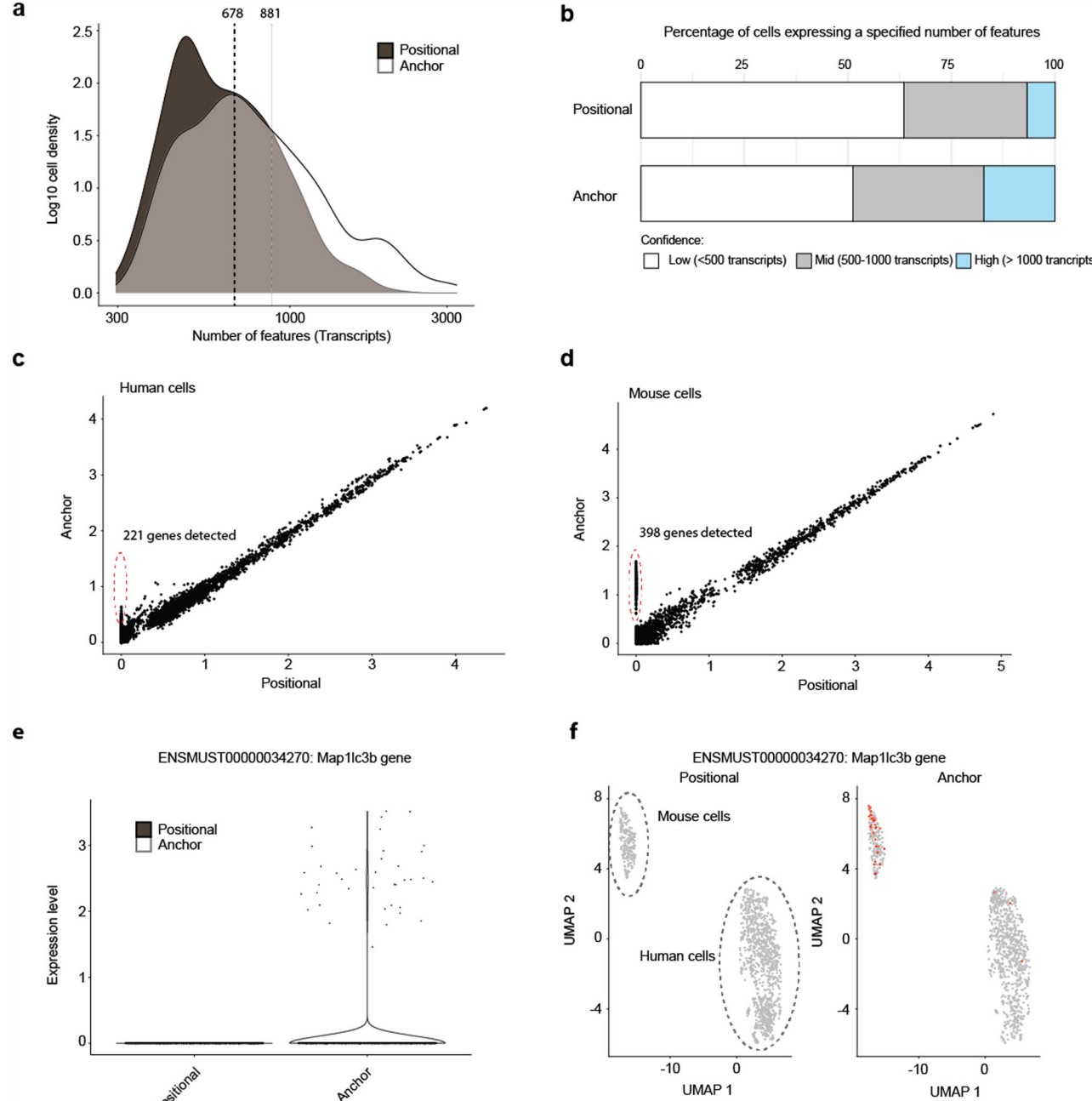

**Fig. 5 | Including an anchor into the beads increases the number of UMIs and transcripts detected. a** The log10 density of transcripts per cell for both positional and anchor UMI selection approaches. **b** The bar plot compares the percentage of cells expressing low ( < 500 transcripts), mid (500-1000 transcripts), and high ( > 1000 transcripts) numbers of features between the positional and anchor UMI selection approaches. The confidence categories for transcript counts are indicated by colour coding: low (white), mid (grey), and high (blue). Data is representative of one of three independent experiments. **c** Correlation of transcripts in human-origin cells between the positional and anchor approaches. Transcripts with enhanced detection using the anchor method are highlighted with a red circle. **d** Correlation of mouse-origin transcripts between the positional and anchor approach, with the red circle identifying transcripts more readily detected using the anchor UMI identification approach. **e** Expression profile of the transcript ENSMUST00000034270 for the positional and anchor approach. **f** UMAP plots showing the increased expression of ENSMUST00000034270 using the anchor approach. Each dot represents a single cell. The data shown is from one of three independent experiments. For plots (**c–f**), each dot represents a single cell. The data shown is from one of three independent experiments.

5'-Bead-HEG_Linker-TCTCTCTCTACACGACGCTCTTCCGATC TJJJJJJJJJJJJBAGCNNNNNNNNNVTTTTTTTTTTTTTTTTTTTTTTTTTT-TTTTTT-3'

Here, "J" represents the monomer split-and-pool barcode, while "N" signifies the dimer amidite UMI and B signifies either a C, G or T. We procured CMI oligos from Sigma-Aldrich with desalting, designed as follows:

Anchor oligo:

5'-TCTCTCTCTACACGACGCTCTTCCGATCTAGTGCGTAGC TGBAGCGGAACCTTGGCCTTAATTGGTTAAGGTTGGAATTTTTT TTTTTTTTTTT-3'

No anchor:

5'-TCTCTCTCTACACGACGCTCTTCCGATCTAGTGCGTAGC TGGGAACCTTGGCCTTAATTGGTTAAGGTTGGAATTTTTTTTTTT TTTTTT-3'

https://doi.org/10.1038/s42003-025-07474-5 **Article**

## Bead sequencing library preparation

Two-thousand beads (in TE/TW storage buffer) were added to a PCR tube. Beads were washed in 200 µl of H2O, centrifuged (100 × g, 1 min) and supernatant removed. 50 µl of PCR mix (1.5 µl indexed polyA primer (100 µM), 1.5 µl new P5 primer (100 µM; for beads carrying a SMART PCR handle) or 1.5 µl NEBNext i50x primer (100 µM; for beads carrying P5 PCR handle) or 10 µl SI primer (from 10x kit; for 10x beads), 25 µl KAPA HiFi master mix and H2O to 50 µl) was added, mixed and immediately run in a thermocycler with the following conditions. 95 °C for 3 min. 12 cycles of 95 °C for 20 s, 60 °C for 15 s, 72 °C for 15 s, then 72 °C for 1 min and 4 °C hold. After PCR, samples were cleaned up by adding 100 µl of SPRIselect beads (Beckman Coulter) and following the manufacturer's instructions. Samples were eluted in 20 µl of H2O and run on an HS D1000 tape (Agilent).

ONT Flongle libraries were prepared and loaded according to protocol sqk-lsk114-ACDE_9163_v114_revJ_29Jun2022-flongle, starting with 100 fmol of PCR product (or pooled products where indexing was used). 10 fmol of library was loaded onto a flongle flow cell. Sequencing was run for 24 h using super-accuracy basecalling, minimum fragment size of 20 bp and no filtering based on Q score. Primers used for the PCR and sequencing are listed in Table 1.

## 10x chromium library preparation

We prepared a single-cell suspension using Jurket and 5TGM1 cells using the standard 10x Genomics chromium protocol as per the manufacturer's instructions. Briefly, cells were filtered into a single-cell suspension using a 40 µM Flomi cell strainer before being counted. We performed 10x Chromium library preparation following the manufacturer's protocol. Briefly, we loaded 3300 Jurkat:5TGM1 cells at a 70:30 split into a single channel of the 10x Chromium instrument. Cells were barcoded and reverse transcribed into cDNA using the Chromium Single Cell 3′ library kit and get bead v3.1. We performed 10 cycles of PCR amplification before cleaning up the library using 0.6x SPRI Select beads. The library was split, and a further 20 or 25 PCR cycles were performed using a biotin oligonucleotide (5-PCBioCTA-CACGACGCTCTTCCGATCT), and then cDNA was enriched using DynabeadsTM MyOneTM streptavidin T1 magnetic beads (Invitrogen). The beads were washed in 2x binding buffer (10 mM Tric-HCL ph7.5, 1 mM EDTA and 2 M NaCl) then samples were added to an equi-volume amount of 2x binding buffer and incubated at room temperature for 10 min. Beads were placed in a magnetic rack and then washed twice with 1x binding buffer. The beads were resuspended in H2O and incubated at room temperature, and subjected to long-wave UV light ( ~ 366 nm) for 10 min. Magnetic beads were removed, and the library was quantified using the QubitTM High sensitivity kit. Libraries were then prepared before sequencing.

## Drop-seq and scCOLOR-seqv2 library preparation

Single-cell capture and reverse transcription were performed as previously described[5,11,26]. Briefly, Jurkat and 5TGM1 cells (20:80 ratio) were filtered

into a single-cell suspension using a 40 µM Flomi cell strainer before being counted. Cells were loaded into the DolomiteBio Nadia Innovate system at a concentration of 310 cells per µL. Custom synthesised beads were loaded into the microfluidic cartridge at a concentration of 620,000 beads per mL. Cell capture was then performed using the standard Nadia Innovate protocol according to manufacturer's instructions. The droplet emulsion was then incubated for 10 min before being disrupted with 1H,1H,2H,2H-per-fluoro-1-octanol (Sigma) and beads were released into an aqueous solution. After several washes, the beads were subjected to reverse transcription. Prior to PCR amplification, beads were treated with ExoI exonuclease for 45 min. PCR amplification was then performed using the SMART PCR primer (AAGCAGTGGTATCAACGCAGAGT) and cDNA was subsequently purified using AMPure beads (Beckman Coulter). The library was split and a further 20 or 25 PCR cycles were performed using a biotin oligonucleotide (5—PCBioTACACGACGCTCTTCCGATCT) and then cDNA was enriched using DynabeadsTM MyOneTM streptavidin T1 magnetic beads (Invitrogen). The beads were washed in 2x binding buffer (10 mM Tric-HCL ph7.5, 1 mM EDTA and 2 M NaCl), then samples were added to an equi-volume amount of 2x binding buffer and incubated at room temperature for 10 min. Beads were placed in a magnetic rack and then washed twice with 1x binding buffer. The beads were resuspended in H2O and incubated at room temperature, and subjected to long-wave UV light ( ~ 366 nm) for 10 min. Magnetic beads were removed, and the library was quantified using the QubitTM High sensitivity kit. Libraries were then prepared for sequencing.

## Single-cell Nanopore library preparation for sequencing

A total of 500 ng of single-cell PCR input was used as a template for ONT library preparation. Library preparation was performed using the SQK-LSK114 (kit V14) ligation sequencing kit, following the manufacturer's protocol. Samples were then sequenced on either a Flongle™ device or a PromethION™ device using R10.4 (FLO-PRO114M) flow cells. The fast5 sequencing data was basecalled to fastq files using guppy basecaller (v6.5.7) using the Super-accuracy mode within the MinKnow software (v23.04.6).

## Read simulation

To understand how much improvement in successfully localising the start of UMIs can be gained from our optimised bead design, we conducted a series of in silico experiments by simulating beads at different application scenarios, including PCR/sequencing substitution errors and PCR/sequencing indels (i.e., insertion and deletion errors). Due to the known composition of the proposed bead, we began directly by amplifying one bead at a single genomic locus over a predefined number of PCR cycles. The overall parameters for simulation were set as follows. Reads were PCR amplified over 8 cycles with an amplification efficiency rate of 0.9. Both PCR and sequencing error rates varied from $10^{-5}$ to $10^{-1}$. The PCR and sequencing error rates were fixed to be $10^{-5}$ and $10^{-3}$, respectively, when another parameter needed to vary. As suggested by Potapov et al.[27], we set

## Table 1 | polyA primers

| polyA_1 | CAAGCAGAAGACGGCATACGAGAT**CGTGAT**GTGACTGGAGTTCAGACGTGTGCTCTTCCGATCTAAAAAAAAAAAAAAAAAAAAAAAA |
| polyA_2 | CAAGCAGAAGACGGCATACGAGAT**ACATCG**GTGACTGGAGTTCAGACGTGTGCTCTTCCGATCTAAAAAAAAAAAAAAAAAAAAAAAA |
| polyA_3 | CAAGCAGAAGACGGCATACGAGAT**GCCTAA**GTGACTGGAGTTCAGACGTGTGCTCTTCCGATCTAAAAAAAAAAAAAAAAAAAAAAAA |
| polyA_4 | CAAGCAGAAGACGGCATACGAGAT**TGGTCA**GTGACTGGAGTTCAGACGTGTGCTCTTCCGATCTAAAAAAAAAAAAAAAAAAAAAAAA |
| polyA_5 | CAAGCAGAAGACGGCATACGAGAT**CACTGT**GTGACTGGAGTTCAGACGTGTGCTCTTCCGATCTAAAAAAAAAAAAAAAAAAAAAAAA |
| polyA_6 | CAAGCAGAAGACGGCATACGAGAT**ATTGGC**GTGACTGGAGTTCAGACGTGTGCTCTTCCGATCTAAAAAAAAAAAAAAAAAAAAAAAA |
| **Bead PCR handle primers** | |
| New P5 | AATGATACGGCGACCACCGAGATCTACACGCCTGTCCGCGGAAGCAGTGGTATCAACGCAGAGTAC |
| NEBNext i50x | AATGATACGGCGACCACCGAGATCTACAC**[index]**ACACTCTTTCCCTACACGACGCT CTTCCGATCT |
| SI | AATGATACGGCGACCACCGAGATCTACACTCTTTCCCTACACGACGCTC |

PCR deletion and insertion rates as $2.4 \times 10^{-6}$ and $7.1 \times 10^{-7}$, respectively, but varied them from $10^{-5}$ to $10^{-1}$, so did sequencing deletion and insertion rates. After PCR amplification, 5000 reads were subsampled for sequencing. The numbers of both substitution errors and indels were determined from negative binomial distributions. The positions of the error bases were randomly picked. To minimise the variance, we executed 10 permutation tests for each application scenario.

### Simulation for comparing positional and anchor schemes

To precisely localise the UMI from a read, our proposed anchor strategy involved incorporating an anchor interposed between a barcode and a UMI as well as a V base placed after the UMI for demarcating its followed poly(dT)s. To ascertain the improvement in identifying UMIs by this design, we further implemented a positional scheme for comparison, which supposedly reaches the entrance to the UMI by counting 16 bases from an ending string of 14 bases in the primer. To test the robustness of the two strategies in coping with high error rates, we subjected simulated reads into either PCR amplification or sequencing at error rates of $10^{-5}$, $2.5 \times 10^{-5}$, $5 \times 10^{-5}$, $7.5 \times 10^{-5}$, 0.0001, 0.00025, 0.0005, 0.00075, 0.001, 0.0025, 0.005, 0.0075, 0.01, 0.025, 0.05, 0.075, and 0.1. To evaluate the efficiency, we calculated $P$, defined as

$$P = \frac{n}{N}$$

where $N$ is a constant 5000 for the subsampled reads and $n$ represents the number of reads whose UMIs can successfully be discovered by using the anchor or positional strategy.

### Simulation for UMI identification with or without an anchor

It has been widely considered difficult to correctly extract UMIs from reads that suffer indels. The inclusion of an anchor in between a UMI and a segment of poly(dT)s makes it possible to be protected from being contaminated by reading Ts into the UMI portion, while the inclusion of an anchor in between a cell barcode and a UMI can set a buffer that suffices to separate the UMI from the cell barcode, which leaves room for UMI identification. To quantify the influence of indels on reads, we simulated scCOLOR-seq reads by adding indels during the PCR amplification or sequencing stage at error rates of $10^{-5}$, $2.5 \times 10^{-5}$, $5 \times 10^{-5}$, $7.5 \times 10^{-5}$, 0.0001, 0.00025, 0.0005, 0.00075, 0.001, 0.0025, 0.005, 0.0075, 0.01, 0.025, 0.05, 0.075, and 0.1. To evaluate their efficiency, we constructed a $Q$ value, computed by

$$Q = \frac{m}{N}$$

where $m$ represents the number of reads whose UMIs can successfully be discovered if the anchor (BAGC) is used, and the number of reads that are free from indels (taken as a strategy for evaluation without an anchor), otherwise.

### Characterisation of the quantity of reads captured by successfully identifying their UMIs

A logistic function $l$ was used to characterise how the number of successfully captured reads (i.e., $P$ and $Q$) varies against both substitution errors or indels. It has the following form.

$$l(x) = \frac{a}{1 + e^{-b(x-c)}} + \varepsilon$$

where $a$, $b$, $c$, and $\varepsilon$ are four parameters to be estimated according to the computed $P$ and $Q$ values. Each logistic equation reflects the influence of substitution errors or indels imposed on reads captured by successfully identifying their UMIs and represents the robustness of our proposed and compared methods in UMI identification.

### 10x Genomics datasets of short-reads

To comprehensively understand how genomic positions from sequencing reads are represented by different nucleotides, we conducted a large-scale analysis over 41 scRNA-seq datasets downloaded from 10x Genomics using chemistry v2, v3, and v3.1 (Supplementary Data 1). To make the data representative of universality and diversity, we chose these datasets with a few criteria, such as from a variety of species, including human, mouse, and their mixture, and from cells or nuclei.

### 10x chromium short-read analysis workflow

The data was processed using a custom CGAT-core[28] pipeline 'pipeline_10x_shortread', which is included within the TallyTriN Github repository (https://github.com/cribbslab/TallyTriN/blob/main/tallytrin/pipeline_10x_shortread.py). Briefly, the quality of each fastq file is evaluated using fastqc toolkit and summary statistics collated using Multiqc[29]. We then identify putative barcodes using UMI-tools whitelist module and then extract the barcodes and UMIs from the read 1 fastq file and append them onto the read 2 file using umi_tools extract module. Hisat2[30] is then used to map the reads to the hg38_ensembl98 genome and the resulting bam file is then sorted, and each read is assigned to a feature using featureCounts[31], with the alignment written to the XT flag of the output bam file. This is then indexed using samtools[32] and then UMI counting is performed using the umi_tools count module before being converted to a market matrix format. The resulting matrix files are then parsed into R/Bioconductor (v4.0.3) using the BUSpaRse (v1.14.1) package and downstream analysis was performed using Seurat (v 4.3.0.1). Transcript matrices were cell-level scaled and log-transformed. The top 2000 highly variable genes were then selected based on variance stabilising transformation which was used for principal component analysis (PCA). Clustering was performed within Seurat using the Louvain algorithm. To visualise the single-cell data, we projected data onto a Uniform Manifold Approximation and Projection (UMAP).

### 10x chromium long-read analysis workflow

To analyse the 10x chromium long-read data, we developed a custom cgatcore pipeline named 'pipeline_10x' in the TallyTriN Github repository (https://github.com/cribbslab/TallyTriN/blob/main/tallytrin/pipeline_10x.py). We split the fastq file into segments to optimise processing time. Each read's polyA tail was identified and reverse complemented for consistent orientation, discarding reads without a polyA tail. We identified the barcode and UMI by locating the PCR primer sequence 'AGATCGGAAGAGCGT' through pairwise alignment, and then extracted the 16 bp barcode and 12 bp UMI based on position. Barcodes were corrected using a whitelisting method similar to UMI-tools.

Mapping was done using minimap2 (v2.22) with settings: -ax splice -uf MD –sam-hit-only –junc-bed, referencing the human hg38 and mouse mm10 transcriptomes. The resultant Sam file was arranged and indexed via Samtools. Read counting employed UMI-tools' count module, converting counts to a matrix format. We processed the raw expression matrices using R/Bioconductor (v4.0.3) and devised scripts to depict barnyard plots, displaying mouse and human cell proportions. Matrices were cell-level scaled and centre log ratio transformed. We selected the top 2000 variably expressed genes post-variance stabilising transformation for PCA. Clusters were identified in Seurat using the Louvain algorithm. For visualisation, we projected the single-cell data onto a Uniform Manifold Approximation and Projection (UMAP).

### Drop-seq analysis workflow

Drop-seq data was processed using a CGAT-core workflow 'pipeline_macosko', which is included within the TallyTriN repository (https://github.com/cribbslab/TallyTriN/blob/main/tallytrin/pipeline_singlecell_macosko.py). Briefly, the fastq file was split into chunks so that analysis scripts could be processed on sections of the data to reduce the processing time. The polyA tail for each read was identified and then reverse complemented to keep all the reads within the same orientation, any

reads not containing a polyA tail were discarded. Next, the barcode and UMI were identified based on the identification of the PCR primer sequence using pairwise alignment and then selecting the 12 bp barcode and the 8 bp UMI using positional matching. Next, the barcodes were corrected using a whitelisting approach like the one implemented by UMI-tools. The reads were then merged and then mapping was performed using minimap2[33] (v2.22). Mapping settings we as follows: -ax splice -uf MD –sam-hit-only –junc-bed and using the reference transcriptome for human hg38 and mouse mm10. The resulting sam file was sorted and indexed using Samtools. Mapping settings we as follows: -ax splice -uf MD –sam-hit-only –junc-bed and using the reference transcriptome for human hg38 and mouse mm10. The resulting Sam file was sorted and indexed using Samtools. Counting was performed using UMI-tools count module before being converted to a market matrix format. Raw transcript expression matrices generated were processed using R/ Bioconductor (v4.0.3), and custom scripts were used to generate barnyard plots showing the proportion of mouse and human cells. Transcript matrices were cell-level scaled and centre log ratio transformed. The top 2000 highly variable genes were then selected based on variance stabilising transformation which was used for principal component analysis (PCA). Clustering was performed within Seurat using the Louvain algorithm. To visualise the single-cell data, we projected data onto a Uniform Manifold Approximation and Projection (UMAP).

### scCOLOR-seqv2 analysis workflow

To process the drop-seq data, we wrote a custom cgatcore pipeline (https://github.com/cribbslab/TallyTriN). We followed the workflow previously described for identifying barcodes and UMIs using scCOLOR-seq sequencing analysis[11]. Briefly, to determine the orientation of our reads, we first searched for the presence of a polyA sequence or a polyT sequence. In cases where the polyT was identified, we reverse-complemented the read. We next identified the barcode sequence by searching for the polyA region and flanking regions before and after the barcode. The dimer UMI was identified based on the primer sequence TCTTCCGATCT at the TSO distal end of the read. Barcodes and UMIs that had a length of less than 50 base pairs were discarded. Next, the barcodes were corrected using a whitelisting approach like the one implemented by UMI tools. The reads were then merged and then mapping was performed using minimap2[33] (v2.22). Mapping settings we as follows: -ax splice -uf MD –sam-hit-only –junc-bed and using the reference transcriptome for human hg38 and mouse mm10. The resulting Sam file was sorted and indexed using Samtools. Mapping settings we as follows: -ax splice -uf MD –sam-hit-only –junc-bed and using the reference transcriptome for human hg38 and mouse mm10. The resulting Sam file was sorted and indexed using Samtools. The transcript name was then appended to the bam XT flag using the xt_tag_nano script before umi_tools count module was used to count the features using the following settings: –per-gene –gene-tag=XT –per-cell –dual-nucleotide. The umi_tools used to correct for the dimer UMIs is located in the AC-dualoligo in a fork at the repository: https://github.com/Acribbs/UMI-tools. Downstream analysis in R was then performed as described for the Drop-seq analysis workflow above. A summary of the analysis workflow is shown in Supplementary Fig. 11.

### Statistics and reproducibility

All experiments were performed with a minimum of three biological replicates unless stated otherwise. Statistical analyses were conducted using R (v4.0.3) and GraphPad Prism (v10.3.0). For comparisons between two groups, unpaired two-tailed t-tests were applied, assuming equal variance unless otherwise noted.

Error bars in figures represent the standard deviation (SD) across replicates unless otherwise specified. For sequencing-based experiments, all datasets were processed using standardised pipelines described in the "Methods" section, and batch effects were controlled using Seurat's integration workflow. The identification of highly variable genes and

dimensional reduction (e.g., PCA, UMAP) was consistent across all single-cell experiments, ensuring reproducibility.

Simulated datasets used for evaluating UMI and barcode recovery rates were generated with predefined error rates (PCR and sequencing error rates of 10^-5 to 10^-1). Logistic regression models were used to fit the proportion of recovered reads (P and Q values) under varying error conditions, and confidence intervals for regression fits were computed to assess the robustness of the anchor-based UMI identification strategy.

For species-mixing experiments, the ratio of human-to-mouse cells was confirmed by barnyard plots derived from mapped reads. Differential expression analyses were performed using the Wilcoxon rank-sum test in Seurat. UMI and feature counts were validated across both short- and long-read sequencing datasets to ensure consistency.

All code used in this study is openly available on GitHub (https://github.com/cribbslab/TallyTriN), and raw sequencing data has been deposited in the GEO repository under accession number GSE263458. All experimental and computational procedures adhere to the guidelines set forth for reproducibility in single-cell RNA-sequencing studies.

### Data availability

Source data is provided in this manuscript. Sequencing data have been deposited in the GEO under accession number GSE263458. Public Drop-seq data was obtained using the GEO accession number: GSE63473. All other 10x datasets were downloaded publicly from the 10x datasets on their website (https://www.10xgenomics.com/datasets), and each dataset is described in Supplementary Data 1—Tables 2–4. Data analysis in Fig 1b, c was from dataset "10k_hgmm_3p_nextgem_Chromium_Controller".

### Code availability

All custom pipelines used within this analysis are available on Git Hub (https://github.com/cribbslab/TallyTriN).

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

## Acknowledgements

Research support was obtained from Innovate UK (T.B. Sr, T.B. Jr, U.O., M.P., A.P.C.), the Engineering and Physical Sciences Research Council (U.O., M.P., A.P.C.), the National Institute for Health Research Oxford Biomedical Research Unit (U.O.), Cancer Research UK (U.O. and A.P.C.), the Bone Cancer Research Trust (A.P.C. and U.O.), the Leducq Epigenetics of Atherosclerosis Network programme grant from the Leducq Foundation (U.O.), the Chan Zuckerberg Initiative (A.P.C.) and the Myeloma Single-Cell Consortium (U.O.). A.P.C. is a recipient of a Medical Research Council career development fellowship (grant no. MR/V010182/1).

## Author contributions

A.P.C. designed the study with contributions from M.P., T.B. Jr, T.B. Sr and U.O. A.P.C. and J.S conducted data analysis, generated the figures and wrote the paper with input from all authors. M.P., D.L., S.H., G.H., J.R., N.M., E.C. and V.G. performed experiments.

## Competing interests

M.P., U.O. and A.P.C. are inventors on patents filed by Oxford University Innovations for single-cell technologies and are co-founders of Caeruleus Genomics. T.B. Jr. is a director and shareholder of ATDBio. T.B. Sr. is a consultant to ATDBio. The other authors declare no competing interests.
