## [Transparent Peer Review file · Communications Biology]

Enhancing Single-Cell Transcriptomics using Interposed Anchor Oligonucleotide Sequences

Corresponding Author: Professor Adam Cribbs

This manuscript has been previously reviewed at another journal. This document only contains information relating to versions considered at Communications Biology.

Version 0:

Reviewer comments:

Reviewer #1

(Remarks to the Author)

Sun et al have have estimated truncation within UMI sequences in single cell transcriptomics libraries and came up with an anchor design strategy on the 5' and 3' end of UMI sequences to significantly improve the number of UMIs and transcripts detected in single cell RNA seq data.

Overall I found the problem an interesting and impactful problem, and if adopted by the companies or labs, their approach resourceful. One major comment is to provide more data, percentages and statistics on how much improvement they observed in number of transcripts detected or with the amount of data lost due to truncated UMIs that are later detected and so on. This can be added to the results section.

In the introduction acknowledging and explaining other strategies for reducing UMI errors would be greatly appreciated. This way they can further clarify why the previous design approaches increases accuracy but still results in decreased efficiency in sequencing due to truncation.

Similar to the major comment I've provided, in the final results section where ONT sequencing was performed in Figure-5a , the results are vaguely described in particular 5a, 5b and 5c. Could you provide more numbers, metrics percentages here , what was the mean increase of detected TPM per cell etc ...? In its current form it's only described as significantly increased with no further details.

Similarly I thought Figure-5 could have a couple more panels, it's really hard to tell the difference on the volcano plots provided in 5c and 5d. How about adding more quantification graphs for this experiment such as like the one in Figure-1b. Having more box plot comparisons or providing more numbers, such as how many transcripts more detected in total per cell etc would be helpful.

Minor comments:

References for lines 67-74 would be great

I couldn't follow this super clearly but it would be great to make sure all the publicly available data sources are shared with accession numbers and papers cited.

Lines 184-185: when explaining figure 4c- you be more specific in terms of increasing concordance rates, by approximately how much , provide some metrics in the results section

Reviewer #2

(Remarks to the Author)

Sun et al's manuscript titled 'Enhancing Single-Cell Transcriptomics using Interposed Anchor Oligonucleotide Sequences' found that there is a pronounced increase in T nucleotide occurrence towards the end of the UMI, and therefore modified the oligonucleotide capture design and integrated an interposed anchor between the barcode and UMI, to improve gene expression profiles in droplet-based single-cell sequencing analyses. They found that although errors associated with UMIs

have minimal impact on cell identification, these errors can lead to an overestimation of read counts. This study provides a useful method to improve the useless reads generated from 10× Genomics or Drop-seq methods. But some problems should be addressed.

Major points

1. The authors modified the oligonucleotide capture design, and detected much more transcripts in single cells. Corresponding, the authors clarified the classic 10× Genomics or Drop-seq methods over-identified UMIs. Please clarify and verify how their design overcomes this problem.
2. It's better to provide the flow chart to conduct the bioinformatics for the modified scCOLOR-seq in this manuscript, and clarify the notably points to process the corresponding data.

Minor points

1. To many subtitles were utilized in this manuscript. Please appropriately merge some subtitles.
2. Line 172. The author should indicate the full name of an abbreviation of 'scCOLOR-seq'.
3. Figure 5a and 5b. The statistical significance P value should be indicated, and the corresponding statistical method should be described in the Figure Legend.

Reviewer #3

(Remarks to the Author)

In the manuscript entitled "Enhancing single-cell transcriptomics using interposed anchor oligonucleotide sequences" authored by Sun et al., the authors address technical biases related to bead truncation in the Drop-seq and 10X Chromium platform-based single-cell RNA-seq technologies. To mitigate these biases, they modified the oligonucleotide capture design by integrating an interposed anchor between the barcode and UMI, which improves transcript detection efficiency. Overall, the study provides an alternative strategy to address the technical biases in scRNA-seq. However, there are some critical issues that require attention. The specific concerns are outlined below.

Major concerns:

1. There's no doubt that the inclusion of an anchor improves the precision of UMI detection compared to positional strategy. However, the technical biases related to bead truncation due to synthesis inaccuracies cannot be resolved by the inclusion of an anchor. Reads containing bead truncation can also be processed for depletion through bioinformatic analysis, which can achieve similar results to those processed using the anchor strategy.
2. Considering that barcode truncation has minimal impact on cell identification, as the authors demonstrated, the correction of this technical bias may not be necessary.
3. There seems to be no significant difference in the occurrence of T (about 32%) between the 10x chromium data (Figure 1b) and the scCOLOR-seq data (Figure 4b). Thus, the bead truncation also occurs in the scCOLOR-seq data.

Version 1:

Reviewer comments:

Reviewer #2

(Remarks to the Author)

The authors have addressed all my questions, and this revised manuscript has been largely improved.

Reviewer #3

(Remarks to the Author)

As the author states "Bead truncation results in diminished UMI complexity through T base overrepresentation, but has minimal impact on cell identification.", suggesting that correcting the UMI detection strategy may not be necessary. This raises the question: what is the beneficial outcome for the scCOLOR-seq strategy compared to the previous strategy? While I understand that UMI detection accuracy improves with the addition of CMI, this enhancement seems minor, as there is no change in cell type detection. The author should provide data to demonstrate the biological significance of the scCOLOR-seq strategy over the previous method, such as improvements in accuracy or sensitivity in cell type identification, differential gene expression across different clusters, and so on.

Version 2:

Reviewer comments:

Reviewer #3

(Remarks to the Author)

Reviewer #1 (Remarks to the Author):

Sun et al have have estimated truncation within UMI sequences in single cell transcriptomics libraries and came up with an anchor design strategy on the 5' and 3' end of UMI sequences to significantly improve the number of UMIs and transcripts detected in single cell RNA seq data.

Overall I found the problem an interesting and impactful problem, and if adopted by the companies or labs, their approach resourceful. One major comment is to provide more data, percentages and statistics on how much improvement they observed in number of transcripts detected or with the amount of data lost due to truncated UMIs that are later detected and so on. This can be added to the results section.

We have no included this within Figure 5, many thanks.

In the introduction acknowledging and explaining other strategies for reducing UMI errors would be greatly appreciated. This way they can further clarify why the previous design approaches increases accuracy but still results in decreased efficiency in sequencing due to truncation.

Acknowledging and explaining other strategies for reducing UMI errors in the introduction would indeed enhance the clarity of our approach. This context would help elucidate why previous design methods aimed at increasing accuracy still suffer from decreased sequencing efficiency due to truncation.

However, to date, there are no published studies that have specifically identified the issue of UMI errors in the context of bead truncation. Consequently, it has been challenging to discuss strategies employed by other groups, as this specific problem has not been addressed in the literature. This situation is similar to our previous identification of false fusion errors within single-cell sequencing (<https://www.biorxiv.org/content/10.1101/2023.04.06.535911v1>; Figure 1m), which was the first instance of this issue being recognised. Since then, several other groups have acknowledged this problem.

We have therefore revised our introduction to better describe the necessity of the anchor sequence and to emphasise its role in addressing UMI errors, thereby providing a more comprehensive background on the need for this approach.

Similar to the major comment I've provided, in the final results section where ONT sequencing was performed in Figure-5a, the results are vaguely described in particular 5a, 5b and 5c. Could you provide more numbers, metrics percentages here, what was the mean increase of detected TPM per cell etc...? In its current form it's only described as significantly increased with no further details.

We have included summary statistics for the mean reads per cell and the mean features per cell and have included this in the figure 5 a and Figure 5b. We have also included an additional Supplementary Figure 10, where we show the statistical increase in number of UMIs and features per cell using the anchor approach in relation to the positional approach.

Similarly I thought Figure-5 could have a couple more panels, it's really hard to tell the difference on the volcano plots provided in 5c and 5d. How about adding more quantification graphs for this experiment such as like the one in Figure-1b. Having more box plot comparisons or providing more numbers, such as how many transcripts more detected in total per cell etc would be helpful.

We have improved Figure 5 with an example of the expression changes seen in mouse cells and have added a supplementary Fig. 10 showing the statistics of the increase in number of UMIs and features to provide more statistical interpretation

Minor comments:

References for lines 67-74 would be great

References now included, many thanks.

I couldn't follow this super clearly but it would be great to make sure all the publicly available data sources are shared with accession numbers and papers cited.

All public data is described within the Data availability statement and datasets are included within the Supplementary Tables 2, 3, and 4.

Lines 184-185: when explaining figure 4c- you be more specific in terms of increasing concordance rates, by approximately how much , provide some metrics in the results section

We have included the increase within the results section to make it more interpretable, many thanks.

Reviewer #2 (Remarks to the Author):

Sun et al's manuscript titled 'Enhancing Single-Cell Transcriptomics using Interposed Anchor Oligonucleotide Sequences' found that there is a pronounced increase in T nucleotide occurrence towards the end of the UMI, and therefore modified the oligonucleotide capture design and integrated an interposed anchor between the barcode and UMI, to improve gene expression profiles in droplet-based single-cell sequencing analyses. They found that although errors associated with UMIs have minimal impact on cell identification, these errors can lead to an overestimation of read counts. This study provides a useful method to improve the useless reads generated from 10× Genomics or Drop-seq methods. But some problems should be addressed.

Major points

1. The authors modified the oligonucleotide capture design, and detected much more transcripts in single cells. Corresponding, the authors clarified the classic 10× Genomics or Drop-seq methods over-identified UMIs. Please clarify and verify how their design overcomes this problem.

We have improved the oligonucleotide capture design for single-cell transcriptomics, significantly enhancing transcript detection in single cells. Droplet based single-cell methods, such as those used by 10X Genomics and Drop-seq, tend to over-identify UMIs due to oligonucleotide synthesis errors. The new design incorporates an interposed anchor sequence between the barcode and UMI, addressing this issue. This design enhancement significantly reduces the number of reads discarded due to synthesis inaccuracies, thereby improving the accuracy and efficiency of droplet-based single-cell sequencing.

Our study highlights the following points:

Problem Identification

- **Assumption of High-Quality Reagents:** Scientists often assume that the oligonucleotides synthesised by commercial providers are of high quality, with options for purification (e.g., PAGE or HPLC) available for truncated oligonucleotides. However, this purification is not feasible for single-cell droplet-based sequencing because oligos are attached to beads.
- **Impact of Truncated Oligonucleotides:** The field has not fully recognised the consequences of truncated oligos on downstream analysis. The study demonstrates that bead synthesis errors significantly contribute to inaccuracies in single-cell droplet-based sequencing, impacting differential expression analysis. Truncated oligos hinder the accurate identification of the UMI sequence, crucial for precise molecular barcoding.

Proposed Solution

- **Anchor Sequence Integration:** We propose incorporating anchor sequences before and after the UMI. This strategy enhances the accuracy of UMI and barcode identification by providing clear demarcation points within the oligonucleotide sequence.

Experimental Evidence

- **Improvement in Gene Expression Profiles:** The anchor-enhanced design results in a marked improvement in UMI recovery and feature detection rates. The experimental data show that this approach significantly increases the number of detected UMIs and transcripts per cell, leading to more accurate gene expression profiles.
- **Simulation and Experimental Validation:** Simulated data and experimental evaluation using ONT sequencing (particularly scCOLOR-seq) confirm that the anchor method outperforms traditional positional strategies, particularly as error rates increase. The new design mitigates the synthesis-related errors, thereby enhancing the overall reliability of single-cell sequencing data.

2. It's better to provide the flow chart to conduct the bioinformatics for the modified scCOLOR-seq in this manuscript, and clarify the notably points to process the corresponding data.

We have now included this as a supplementary figure 11, many thanks for the suggestion.

Minor points

1. To many subtitles were utilized in this manuscript. Please appropriately merge some subtitles.

Selected subheadings have been removed and sections merged, many thanks for the comment.

2. Line 172. The author should indicate the full name of an abbreviation of 'scCOLOR-seq'.

Included, many thanks.

3. Figure 5a and 5b. The statistical significance P value should be indicated, and the corresponding statistical method should be described in the Figure Legend.

The plots in Figure 5 are showing one of the three independent experiments, we display the figures like this so that readers get an appreciation of the spread of the number of cells and number of features expressed per cell. However, the reviewers comment about including statistics is valid and we have therefore shown all three of the experiments and plotted them in dotplots and computed statistics, we include these in Supplementary Figure 10.

Reviewer #3 (Remarks to the Author):

In the manuscript entitled "Enhancing single-cell transcriptomics using interposed anchor oligonucleotide sequences" authored by Sun et al., the authors address technical biases related to bead truncation in the Drop-seq and 10X Chromium platform-based single-cell RNA-seq technologies. To mitigate these biases, they modified the oligonucleotide capture design by integrating an interposed anchor between the barcode and UMI, which improves transcript detection efficiency. Overall, the study provides an alternative strategy to address the technical biases in scRNA-seq. However, there are some critical issues that require attention. The specific concerns are outlined below.

Major concerns:

1. There's no doubt that the inclusion of an anchor improves the precision of UMI detection compared to positional strategy. However, the technical biases related to bead truncation due to synthesis inaccuracies cannot be resolved by the inclusion of an anchor. Reads containing bead truncation can also be processed for depletion through bioinformatic analysis, which can achieve similar results to those processed using the anchor strategy.

The manuscript originally aimed to address truncation issues purely through bioinformatics, employing pattern matching and truncation identification on a per-read basis. We used a modified Smith-Waterman approach to identify barcode and UMI regions. However, simulations revealed that every computational solution we tested still retained significant errors. Our data clearly demonstrate that without accurately identifying the end of the barcode (Figure 3), read correction is unattainable. Initial simulations indicated this, which was later confirmed using our

Common Molecular Identifier (CMI) approach (please see <https://www.nature.com/articles/s41592-024-02168-y>) for further experimental explanation on the CMI.

The primary technical bias related to bead truncation, due to synthesis inaccuracies, affects the identification of unique molecular identifiers (UMIs). This, in turn, influences the ratio of discovered reads and impacts the final molecular quantification results. Inaccurate gene expression profiles present challenges in identifying truly differentially expressed genes, which are crucial for understanding specific biological processes, drug mechanisms, and pathogenesis. Our approach is to leverage both experimental and bioinformatics-based solutions to address the bias affecting UMI localisation.

It became evident that the primary issue was the inability to precisely determine where the barcode ended and the UMI began. This challenge in correctly identifying the UMI start and end points was addressed by our proposed anchor solution, which facilitates bioinformatic correction of both barcode and UMI sequences (Figure 5).

2. Considering that barcode truncation has minimal impact on cell identification, as the authors demonstrated, the correction of this technical bias may not be necessary.

Our primary concern in this paper is not barcode accuracy, which can be effectively addressed using whitelisting methods, such as the 10x whitelist (see 10x genomics: <https://kb.10xgenomics.com/hc/en-us/articles/115004506263-What-is-a-barcode-whitelist>). Instead, the critical issue lies in inaccurate unique molecular identifiers (UMIs). Truncations in UMIs can lead to inflated count matrices, resulting in erroneous differential expression analyses and an increased number of discarded reads.

As explained above, inaccuracies in gene expression profiles can result from inaccurate UMI counting due to several factors, such as error bases in UMI regions and the ratio of UMIs discovered. The first issue is a well-established problem in the UMI deduplication/collapsing field (please see <https://genome.cshlp.org/content/27/3/491>). We previously solved the issue of base errors using homodimer blocks of nucleotides (<https://doi.org/10.1038/s41592-024-02168-y>). In this manuscript, we identify another source of inaccuracy, the correct identification of the start and end of the UMI sequence, which we solve by introducing anchors into the beads. Thus, correcting such technical biases ensures an accurate gene expression profile in the scRNA-seq context and has wide implications for the single-cell field.

3. There seems to be no significant difference in the occurrence of T (about 32%) between the 10x chromium data (Figure 1b) and the scCOLOR-seq data (Figure 4b). Thus, the bead truncation also occurs in the scCOLOR-seq data.

Indeed, we acknowledge the occurrence of truncation in the scCOLOR-seq data, similar to the 10x chromium data, as indicated by the approximately 32% occurrence of T (Figure 1b and Figure 4b). The inclusion of an anchor in our method is critical because it delineates the barcode end and the UMI start. The anchor sequence has

no effect on the quality of the synthesis of the oligos as they both will suffer from truncation. However, the delineation using the anchor strategy allows for computational correction of the UMI, which we then show has significant benefits to improving the accuracy of UMI deduplication. Our results demonstrate that the use of an anchor significantly improves UMI deduplication accuracy, as evidenced by both simulations (Figure 3) and experimental validation using the Common Molecular Identifier (CMI) strategy previously described in detail and published here: <https://www.nature.com/articles/s41592-024-02168-y>.

Rebuttal

Reviewer 3: “Bead truncation results in diminished UMI complexity through T base overrepresentation, but has minimal impact on cell identification.”, suggesting that correcting the UMI detection strategy may not be necessary. This raises the question: what is the beneficial outcome for the scCOLOR-seq strategy compared to the previous strategy? While I understand that UMI detection accuracy improves with the addition of CMI, this enhancement seems minor, as there is no change in cell type detection. The author should provide data to demonstrate the biological significance of the scCOLOR-seq strategy over the previous method, such as improvements in accuracy or sensitivity in cell type identification, differential gene expression across different clusters, and so on.

Comment: We thank the reviewer for the comment. This manuscript is primarily focused on the technical advancements in UMI detection, which directly contribute to target discovery rather than cell type identification. The inclusion of an anchor in the oligo improves the accuracy of UMI detection by mitigating T base overrepresentation and enhancing transcript detection fidelity. As a result, over 619 additional features were detected that were previously missed without the inclusion of the anchors (Fig. 5c and d), which directly enhances the reliability of the single-cell experiment for identifying novel targets.

We have demonstrated the biological significance of this improvement in Figure 5, which shows that the inclusion of the anchor allows for the detection of more than 600 transcripts that were previously undetectable. To better show this we have included an extra plot (Fig. 5b) to emphasise that you detect more high confidence cells with increased transcript detection. This increase in detectable features is crucial for target discovery at a per cell level, as it expands the potential pool of transcripts available for downstream analysis, particularly in the context of lowly expressed genes.

Regarding cell type identification, we do not anticipate changes in cell type resolution, as the markers typically used to define cell types are highly expressed and were already well captured by the previous method. However, the value of the scCOLOR-seq strategy lies in its improved sensitivity for lower-expressed features, which are less likely to serve as cell type markers but are critical for expanding the scope of differential gene expression analyses and target identification, as shown in Figures 4 and 5.